# Modelling the structures of frameshift-stimulatory pseudoknots from representative bat coronaviruses

**Rohith Vedhthaanth Sekar**[1], **Patricia J. Oliva**[1], **Michael T. Woodside**[1,2,3]*

**1** Department of Physics, University of Alberta, Edmonton, Canada, **2** Li Ka Shing Institute of Virology, University of Alberta, Edmonton, Canada, **3** Centre for Prions and Protein Folding Diseases, University of Alberta, Edmonton, Canada

* michael.woodside@ualberta.ca

**Data Availability Statement:** All relevant data are within the manuscript and its Supporting Information files.

## Abstract

Coronaviruses (CoVs) use −1 programmed ribosomal frameshifting stimulated by RNA pseudoknots in the viral genome to control expression of enzymes essential for replication, making CoV pseudoknots a promising target for anti-coronaviral drugs. Bats represent one of the largest reservoirs of CoVs and are the ultimate source of most CoVs infecting humans, including those causing SARS, MERS, and COVID-19. However, the structures of bat-CoV frameshift-stimulatory pseudoknots remain largely unexplored. Here we use a combination of blind structure prediction followed by all-atom molecular dynamics simulations to model the structures of eight pseudoknots that, together with the SARS-CoV-2 pseudoknot, are representative of the range of pseudoknot sequences in bat CoVs. We find that they all share some key qualitative features with the pseudoknot from SARS-CoV-2, notably the presence of conformers with two distinct fold topologies differing in whether or not the 5′ end of the RNA is threaded through a junction, and similar conformations for stem 1. However, they differed in the number of helices present, with half sharing the 3-helix architecture of the SARS-CoV-2 pseudoknot but two containing 4 helices and two others only 2. These structure models should be helpful for future work studying bat-CoV pseudoknots as potential therapeutic targets.

## Author summary

Structures in coronavirus (CoV) genomes called pseudoknots control expression of viral proteins essential for replication, making them attractive targets for broad-spectrum anti-CoV drugs. However, the 3D structures of most CoV pseudoknots are unknown. Here we computationally model the structures of a set of pseudoknots that are representative of the range of sequences found in bat-CoV pseudoknots, motivated by the fact that bat CoVs are the ultimate source of most human CoV diseases. We find these representative pseudoknots all share some crucial structural features, including very similar configurations of the central helix in the pseudoknot and an unusual topology in which one end of the RNA threads through a ring formed in the structure. The commonality of these features across

**Funding:** Funding was received for this work by MTW from the Canadian Institutes of Health Research (grant reference nos. OV3–170709 and VS1–175534), Alberta Innovates (reference no. G2020000270), the Canada Foundation for Innovation Exceptional Opportunities Fund (grant reference number 40845), and the Li Ka Shing Institute of Virology. The funders had no role in the study design, data collection and analysis, decision to publish, or preparation of the manuscript.

**Competing interests:** The authors have declared that no competing interests exist.

bat-CoV pseudoknots supports the notion that drugs may be able to bind many different CoV pseudoknots, and the rarity of the fold topology suggests such binding should be highly specific to CoVs, implying pseudoknots are well-suited as targets for broad-spectrum anti-CoV drugs. This work provides new insights into CoV pseudoknot structures that may help future efforts targeting pseudoknots therapeutically.

## Introduction

Zoonotic coronaviruses have given rise to multiple public-health emergencies in the last two decades, starting with the SARS epidemic 20 years ago, followed by episodic outbreaks of MERS over the last 10 years, and continuing with the COVID-19 pandemic that began in 2019 [1]. In each case, the novel CoVs causing disease in humans were most closely related to CoVs found in bats [2], one of the largest reservoirs of CoVs [3]. Owing to increasing encroachment by humans into bat habitats, we may expect that bat CoVs will continue to lead to novel zoonotic human CoV diseases in coming years. It is thus urgent to understand more fully the properties of the plethora of CoVs observed in bats, so as to enable the development of broad-spectrum anti-CoV treatments, given that no such treatments currently exist.

A promising possible target for broad-spectrum anti-CoV drugs is −1 programmed ribosomal frameshifting (−1 PRF), a form of translational recoding that plays a crucial role in all CoVs [4]. In −1 PRF, structural elements in the viral RNA induce the ribosome to shift into the −1 reading frame at a programmed location in the genome, leading to the production of an alternate polypeptide [5,6]. In the case of CoVs, −1 PRF is stimulated by a pseudoknot located between ORF1a and ORF1b; the latter, which includes the proteins needed for viral transcription and replication, is only expressed in the −1 frame [4]. Modulating −1 PRF levels by mutating the pseudoknot or binding ligands to it has been shown to attenuate replication of CoVs like SARS-CoV, SARS-CoV-2, and MERS-CoV significantly [7–10]. Similar effects have also been seen for other viruses dependent on −1 PRF such as HIV [11–16]. Recently, compounds found through empirical searches for inhibitors of −1 PRF in specific CoVs like MERS-CoV and SARS-CoV-2 have been shown to be effective as inhibitors of −1 PRF in other human CoVs [9,10,17], and in some cases against −1 PRF in a range of bat CoVs [17], supporting the notion of targeting −1 PRF as a strategy for developing broad-spectrum anti-CoV therapeutics.

However, efforts to discover novel inhibitors effective against a wide range of bat CoVs using structure-based approaches are hampered by the lack of structural information about bat-CoV pseudoknots. The pandemic has stimulated a concerted effort to solve the structure of the SARS-CoV-2 pseudoknot, leading to multiple structural models based on a variety of experimental and computational methods [18–23]. These models have revealed important features, such as the presence of an unusual fold topology featuring the 5′ end of the RNA threaded through a junction between two stems (previously seen only in exoribonuclease-resistant RNAs [24,25]), and the ability of the pseudoknot to take on a variety of different conformations—including different topologies for the 5′ end threading [26–28]. To our knowledge, however, atomistic structural models have not yet been reported for any other CoV pseudoknots. In particular, no information about the structure of pseudoknots from bat CoVs is available.

Here we explore the range of qualitative structural features characteristic of bat-CoV pseudoknots by computational modeling the structures of eight bat-CoV pseudoknots chosen because, together with the SARS-CoV-2 pseudoknot, they are representative of the range of

pseudoknot sequences seen in bat CoVs. Four of these pseudoknots (two each from alpha-CoVs and beta-CoVs) have been confirmed experimentally to stimulate −1 PRF [17]. We first generated blind predictions of 3D structures using FARFAR2 [29,30], based on the likely base-pairing interactions predicted from the sequence using 2D structure-prediction tools designed for RNA pseudoknots. We then used microsecond-long, all-atom molecular dynamics (MD) simulations to assess the stability of the predicted structures and analyze the tertiary interactions defining the most occupied structural clusters. We found that all eight bat-CoV pseudoknots shared the ability of the SARS-CoV-2 pseudoknot to form unusual folds featuring 5′-end threading, with 4 of the 8 also sharing its ability to take on conformers with distinct 5′ end topologies (unthreaded as well as threaded). Whereas all the pseudoknots also shared similar conformations for stem 1, they differed in the number of stems present, which varied from 2 to 4. These results provide a qualitative sense of the range of structures that may be found in pseudoknots from bat CoVs, and suggest some commonalities that might be used to search for broad-spectrum anti-CoV drugs targeting −1 PRF.

## Methods

### Bat-CoV pseudoknot sequence alignment and clustering

We expanded on an earlier effort to identify representative bat-CoV pseudoknots [17] by searching the NCBI virus database [31] for all sequences identified as bat CoVs (taxID 1508220), finding 63 that included full coverage of the frameshift signal. We then used Infernal [32] to align these sequences based on sequence and structural similarities: we constructed a covariance model based on the sequence and SHAPE data [27] for the SARS-CoV-2 frameshift signal (starting from the slippery sequence) using the function 'cmbuild' and calibrated it using the function 'cmcalibrate'. This model was used to search for regions in the bat-CoV sequences homologous to the SARS-CoV-2 frameshift signal using the function 'cmsearch' with -A option, thereby generating a multiple sequence alignment (S1 Data). This alignment was then used to construct a phylogenetic tree based on the Tamura-Nei genetic distance model [33] and the neighbor-joining method [34], using Geneious Prime (version 2022.2.2). The major branches of this phylogenetic tree were identified as representing clusters of pseudoknots with similar sequences and structures.

### Blind structure prediction

Base-paired 2D structures were predicted from the RNA sequence for each of the selected pseudoknots using pKiss [35], HotKnots [36,37], PKNOTS [38], and NUPACK [39]. Predictions were blind in the sense that there was no prior knowledge about structures from experiments that could be used to inform the predictions. We selected the lowest-energy consensus pseudoknotted structure from the predictions for further analysis, choosing the structure returned by the majority of the algorithms used in the case of competing pseudoknot predictions (S1 Table and S2 Data). To generate 3D structure predictions, we used the 2D predictions as input for FARFAR2 [29,30]. FARFAR2 was found previously [20] to provide predictions for the structure of the SARS-CoV-2 pseudoknot that tended to be stable in MD simulations, were similar to structures solved experimentally [18,21–23], and were consistent with single-molecule force spectroscopy [26]. Each RNA sequence was extended at the 5′ end by 3–4 nucleotides (nts) from the adjacent linker sequence, which is unpaired, to distinguish between threaded and unthreaded 5′ ends. For every sequence, we selected the lowest-energy predicted 5′ end threaded conformer as well as the lowest-energy predicted unthreaded conformer for MD simulation. To ensure that the results were not biased by these choices, we also simulated the 2 next-lowest-energy predicted 5′ end threaded conformers (where available) and analyzed

them as for the lowest-energy structure, focusing on the threaded conformers because the threaded conformer is dominant for SARS-CoV-2 [18,26].

## Molecular dynamics simulations

The structures selected for MD simulations were protonated at pH 7 using Molecular Operating Environment (Chemical Computing Group) and taken as the starting structures in all-atom MD simulations with explicit solvent using AMBER18 [40], similar to previous work [20]. The F99bsc0_chiOL3 force field was used to parameterize the pseudoknots, and solvation was done using optimal point charge water boxes with minimum margins of 12 Å via Amber's tleap module. The solvated systems were first neutralized using sodium ions, then their salinities were adjusted to 0.15M NaCl using Joung-Cheatham monovalent ion parameters [41]. Each pseudoknot model was simulated under two conditions: without $Mg^{2+}$ ions; or with six $Mg^{2+}$ ions placed initially along the backbones of S1 and S2 (in the case of the 2-stem pseudoknots), additionally at the junction between S1 and S3 (in the case of the 3-stem pseudoknots), and also in the junction between S1 and S4 (in the case of the 4-stem pseudoknots). Initial placement of $Mg^{2+}$ ions was done manually using MOE. In each case, the solvated system was energy-minimized and heated to 310 K with restraints of 10 kcal/mol/$Å^2$ on the backbone phosphate atoms, before gradually removing the restraints and subsequently simulating the unrestrained system on graphical processing units for at least 1 μs at constant pressure.

## Analysis of MD simulations

Analysis was done using the CPPTRAJ module of AmberTools, similar to previous work [20]. Briefly, the pseudoknot conformations within the last 500 ns of each individual simulation were clustered using mass-weighted root-mean-squared deviation (RMSD) of the non-hydrogen backbone atoms in the structured regions (ignoring the flexible loops, which tended to have large fluctuations), using the hierarchical agglomerative approach. Equilibration during this part of trajectory was tested by confirming that (i) RMSD did not undergo large changes (greater than 3 Å from the average value in the final 500 ns of the trajectory, lasting longer than 20 ns), and (ii) the top clusters recurred repeatedly during the trajectory (shown in RMSD graphs for each trajectory colorized by cluster occupancy); the top 3 clusters occupied 70–100% of the trajectory in every case, further supporting convergence of the simulations. Representative structures of the three most populated clusters of each model (PDB files provided in S3 Data) were extracted from the equilibrated portion of individual trajectories, similar to previous work [20]. Interactions between the nucleotides were inspected on PyMol-Open-Source by viewing hydrogen bonds within the pseudoknot with maximum length of 3.2Å; no angular criteria were used. Tertiary contacts were classified as per the notation of Leontis et al. [42]. The root-mean-squared fluctuation (RMSF) of each residue was also calculated.

## Results

### Bat-CoV pseudoknot sequence alignment and structure prediction

We began by generating a multiple sequence alignment (MSA) of the 63 sequences from the NCBI Virus database [31] listed as bat CoVs that had complete coverage of the frameshift signal region as of Dec 2022. We used Infernal [32] to align these sequences based on sequence and structural similarities, with the SARS-CoV-2 frameshift signal as a reference (S1 Data). Note that because Infernal cannot process pseudoknotted base-pairs, the alignment used non-pseudoknotted base-pairs only. We then used this MSA to construct a phylogenetic tree based

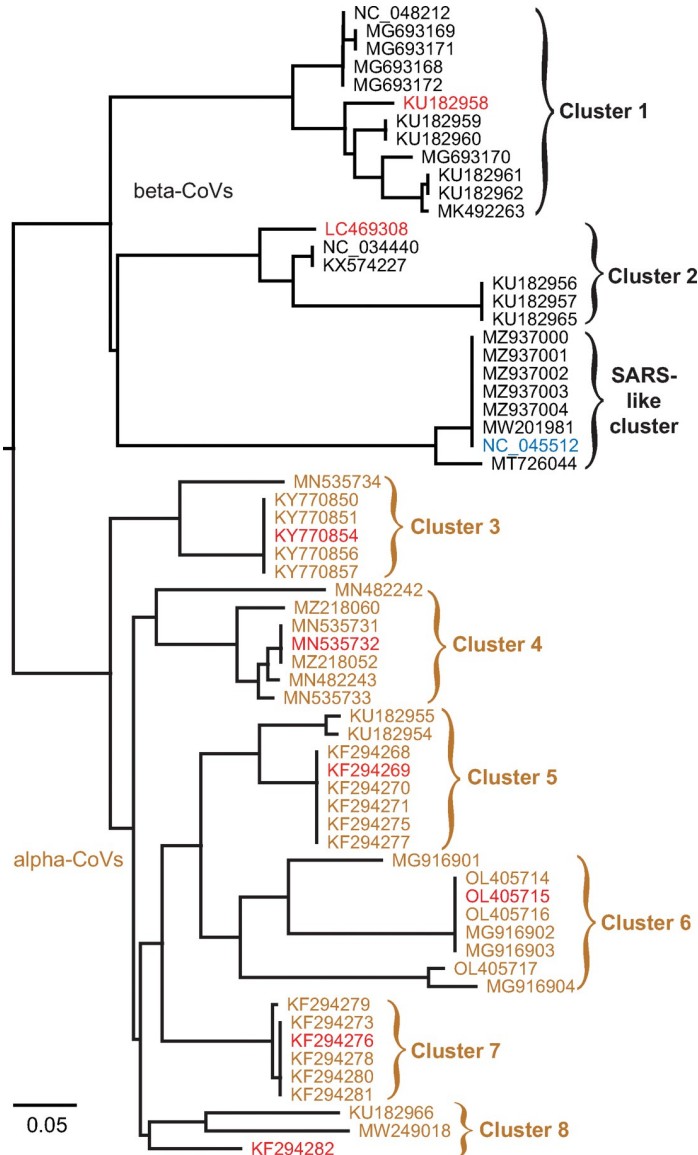

**Fig 1. Bat-CoV pseudoknot clusters.** A phylogenetic tree of bat-CoV frameshift signals identifies 9 clusters with similar pseudoknot sequences and structures. Bat CoVs identified by accession number. Black: beta-CoVs; brown: alpha-CoVs. Sequences chosen for study shown in red. Scale bar: substitutions per site.

on genetic distance (Fig 1). This phylogenetic tree included 9 major branches that we identified as distinct clusters of bat-CoV pseudoknots having similar sequence and structure, 3 in the beta-CoVs (Fig 1, black) and 6 in the alpha-CoVs (Fig 1, brown). One of the beta-CoV clusters was closely related to the pseudoknot sequences from SARS-CoV and SARS-CoV-2, hence it was excluded from analysis because of the significant structural information already available for the SARS-CoV-2 pseudoknot. One representative from each of the remaining clusters was then chosen for structural modelling (Fig 1, red), denoted by the virus isolate name:

- For cluster 1, GLGC2 (beta-CoV from the fruit bat *Rousettus leschenaultia*, NCBI accession number KU182958).

- For cluster 2, Vs-CoV-1 (beta-CoV from the vesper bat *Vespertilio sinensis*, accession number LC469308).

- For cluster 3, Anlong-44 (alpha-CoV from the horseshoe bat *Rhinolophus macrotis*, accession number KY770854).

- For cluster 4, BtCoV/21164-6-alt/M.dau/DK/2015, hereafter abbreviated as 21164–6 (alpha-CoV from Daubenton's bat *Myotis daubentonii*, accession number MN535732).

- For cluster 5, Anlong-8 (alpha-CoV from the bent-wing bat *Miniopterus schreibersii*, accession number KF294269).

- For cluster 6, BtCoV/Rh/YN2012/Rs4236, hereafter abbreviated as Rs4236 (alpha-CoV from *Rhinolophus sinicus*, accession number OL405715).

- For cluster 7, Anlong-172 (alpha-CoV from *M. schreibersii*, accession number KF294276).

- For cluster 8, Neixiang-64 (alpha-CoV from *M. schreibersii*, accession number KF294282).

We confirmed the reliability of the alignment and clustering by comparing the secondary structure (base-pairing) predicted for each of the pseudoknots in the MSA (S1 Table and S2 Data). For each bat-CoV frameshift signal, the section of the sequence identified as the pseudoknot in the MSA (94–109 nts starting at the 7th nt downstream of the UUUAAAC slippery sequence common to all the bat CoVs) was used as the input for predicting base-pairing interactions by four different algorithms: pKiss [35], HotKnots [36,37], PKNOTS [38], and NUPACK [39]. The slippery sequence and linker region upstream of the pseudoknot were not included in the secondary-structure prediction, because during frameshifting those parts of the sequence are inside the ribosome and hence cannot be structured [6]. The lowest-energy consensus pseudoknotted structure predicted by these algorithms (*i.e.*, the pseudoknot that was predicted by the largest number of algorithms, taking the lowest-energy one in the case of multiple possibilities) was selected; the algorithms involved in selecting the consensus are listed in S1 Table. We found that the predicted secondary structures (shown in dot-bracket notation in S1 Table) featured between 2 and 4 stems, with a 3-stem architecture being the most common (seen in 5 of the 9 pseudoknot clusters). Pseudoknots within a given cluster generally had very similar 2D structures, with only a few exceptions. We included the pseudoknot from SARS-CoV-2 in this analysis to show that the secondary structure prediction agrees with base-pairing observed experimentally [18,21–23] as a test of the reliability of the results. We also applied R-scape [43] to the MSA to evaluate both the alignment and the conservation of the SARS-CoV-2 frameshift signal secondary structure used to build the base-pair co-variance model in Infernal with the sequences in the MSA. We found that the MSA is strongly supported in the R-scape test by the presence of two co-varying base-pairs (S1 Fig), and stem 1 is highly conserved across the alignment but stem 3 is much less so (the analysis can't be done for S2 because Infernal uses non-pseudoknotted base pairs to build covariance models). This result is consistent with the secondary structure predictions in S1 Table, where S1 is similar across all bat-CoV pseudoknots but S3 is quite variable among the alpha-CoV pseudoknots.

For each pseudoknot chosen as representative of a cluster, we made blind predictions of the 3D structure based on the RNA sequence. The consensus pseudoknotted secondary structures predicted above (illustrated in S2 Fig) were used as inputs for Rosetta FARFAR2 [29,30] to predict 3D structures. An additional 3–4 nt upstream of the 5′ end of the pseudoknot was added to the sequence used for 3D structure prediction in each case, to allow threaded and unthreaded 5′ ends to be distinguished in the final structures if present [20]. In all cases, the

base-pairing used for input FARFAR2 was preserved in the 3D structure predictions. Inspecting the top 10 structures predicted for each pseudoknot (S3 and S4 Figs), we removed any predictions that included topological knots (as they are not compatible with genomic RNA through which the ribosome must translocate) or threading of the 3′ end (which is disfavored kinetically [26]). The remaining predictions were found to fall into two categories: those with the 5′ end threaded through a helix junction involving stem 1 (S3 Fig), and those with the 5′ end unthreaded (S4A Fig). Five of the eight pseudoknots (GLGC2, Vs-CoV-1, Rs4236, Anlong-172, and Neixiang-64) were predicted to contain three stems, like the SARS-CoV-2 pseudoknot, but two (Anlong-44, 21164–6) were predicted to contain four stems, and one (Anlong-8) was predicted to contain 2 stems.

## Molecular dynamics simulations

We next tested if the predicted structures were dynamically stable in 1-μs-long all-atom molecular dynamics simulations. Visual inspection suggested that the structural differences between predictions within each of the two categories (threaded or unthreaded) for each pseudoknot were relatively small (consisting of minor reorientations of stems and/or fluctuations in loops), hence we focused on the lowest-energy 3D prediction from each group (Fig 2, also shown in S3 [green boxes] and S4 Figs) for use as the initial structures in MD simulations. Each structure was simulated under two conditions: with monovalent salt (NaCl) only, or with both monovalent salt and $Mg^{2+}$ ions present (initial $Mg^{2+}$ positions shown in S5 Fig). $Mg^{2+}$ was found to stabilize the SARS-CoV-2 pseudoknot significantly, although it was not essential for its folding [26]; it is unclear what role is played by $Mg^{2+}$ for the bat-CoV pseudoknots studied here, hence simulations were done with and without $Mg^{2+}$ ions. We note that the

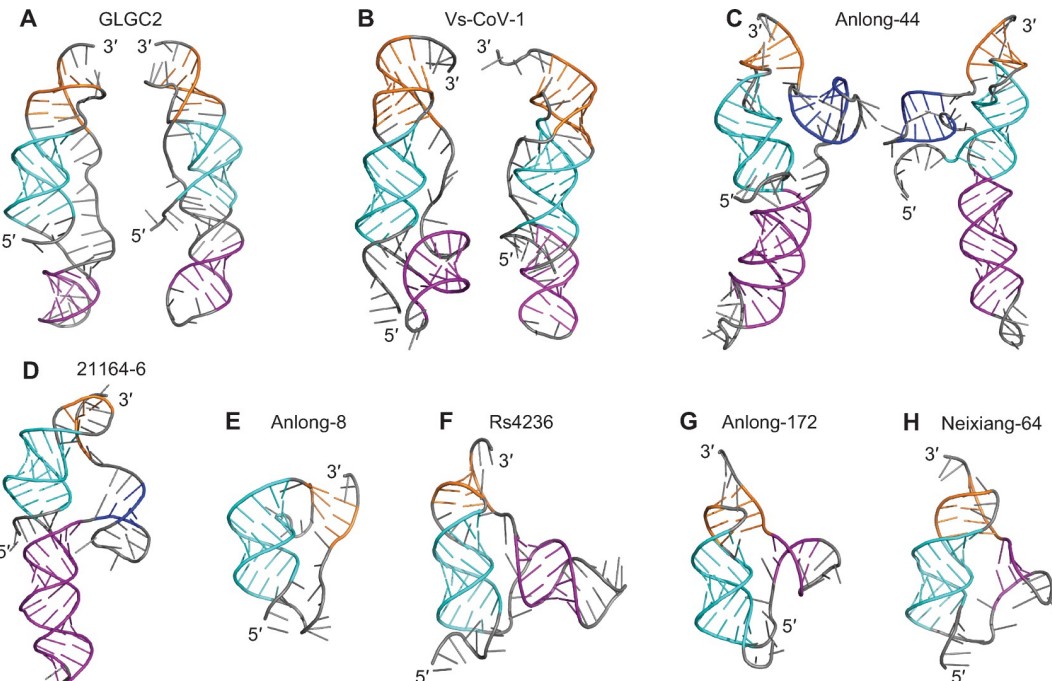

**Fig 2. Blind-prediction structures used to initiate MD simulations.** Lowest-energy pseudoknot structures from FARFAR2. Cyan: S1; gold: S2; magenta: S3; blue: S4; grey: loops or unpaired ends. Unthreaded conformers shown on right in cases where they occur. Base-pairing same as in S2 Fig.

simulations with $Mg^{2+}$ were performed primarily to determine if $Mg^{2+}$ induced gross changes in structure (*e.g.* alternative base-pairing, changes in the number of stems and/or loops), because the lack of information about the number and position of $Mg^{2+}$ ions in the pseudoknots precluded efforts to explore any more detailed effects. The root-mean-square fluctuation in position of the $Mg^{2+}$ ions that remained bound to the pseudoknots during the simulations was at most 7Å.

To test if the results were biased by the choice of initial structure, we also repeated the simulations for the threaded conformers with the second- and third-lowest energies (where they existed), without $Mg^{2+}$. Two-dimensional RMSD plots making pairwise comparisons between the structured regions in the replicate trajectories found no major differences (S6 Fig). For all simulations, we treated the first 500 ns as an equilibration phase, and analyzed the final 500 ns of each simulation to characterize the structures formed by the pseudoknots. Owing to the dynamic nature of the RNA conformations, we clustered the structures by RMSD and analyzed the centroid of the 3 most occupied clusters for each of the pseudoknots. The base pairs identified in each structure were found to be preserved for at least 98.5% of the time in the equilibrated phase of the trajectories.

## Beta-CoV pseudoknots

Considering first the results for the GLGC2 pseudoknot (cluster 1, beta-CoV), we found that its overall structure was quite similar to that of the SARS-CoV-2 pseudoknot (shown for reference in S7 Fig), featuring the same 3-stem architecture where two of the stems interact as in a classic H-type pseudoknot and the third forms as an extended helix in what would otherwise be a large loop (Fig 3). However, none of the stems (denoted S1–S3) featured any bulges, unlike S2 and S3 in the SARS-CoV and SARS-CoV-2 pseudoknots [18–23,44]; such bulges were found to play a significant role in frameshifting in SARS-CoV [44]. The loop connecting S3 to S2 was also much longer than in the SARS-CoV-2 pseudoknot. There was no evidence of

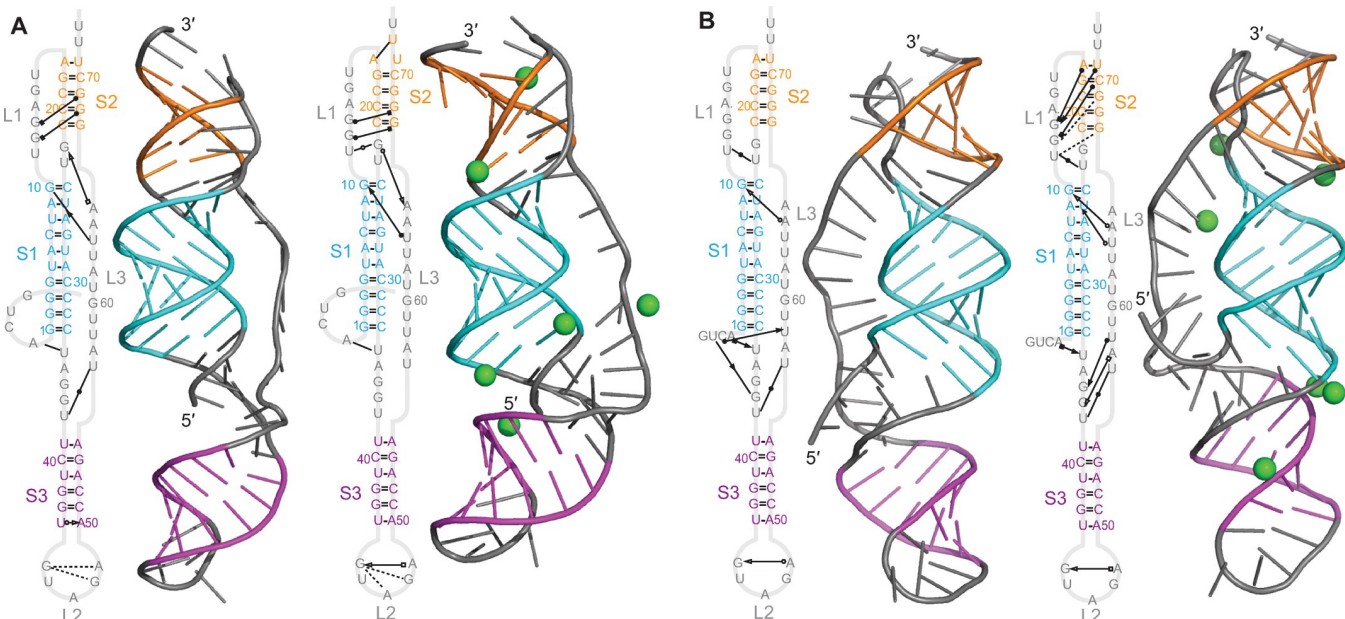

**Fig 3. Structural models for GLGC2 pseudoknot.** (A) Models with 5′ end threaded between S1 and S3, with (right) and without (left) $Mg^{2+}$ (ions shown in green). Secondary structures shown with tertiary interactions notated as in Ref. 38. 3D structure representative of most populated cluster. (B) Models with unthreaded 5′ end.

stacking between S1 and S3; instead, they were separated by an unpaired junction. The base-pairing was unchanged after MD simulations compared to the 2D predictions (S2A Fig). The threaded model (Fig 3A) featured a number of tertiary interactions in the S2/loop 1 (L1) region, S1/L3 region, and the junction between S1 and S2 (as in the SARS-CoV-2 pseudoknot), which did not depend strongly on $Mg^{2+}$. Structural fluctuations were highest in L2, where they were larger than in L3 despite the latter's greater length (S8A Fig). The unthreaded model showed stronger dependence on $Mg^{2+}$, with many tertiary interactions seen in the presence of $Mg^{2+}$ but few without it (Fig 3B). Again, the fluctuations were highest in L2 and lower in L3.

To test replicability, we repeated the simulation of the threaded conformer without $Mg^{2+}$ twice (S9 Fig), and also compared the threaded conformer without $Mg^{2+}$ above to the results of simulations starting from the two next-lowest-energy predictions (S8E Fig). In each case, we saw qualitatively similar results: no base-pairing changes, minor variations in stem orientation (mostly in S3), and broadly similar patterns of tertiary interactions. Some tertiary interactions were constant in all replicates but many or most were not, indicating that while the results are robust with respect to the qualitative structural features, they are less so for details of the tertiary interactions, motivating our qualitative approach to the analysis. Finally, we repeated simulations with $Mg^{2+}$ while placing the ions at different initial locations. Qualitative features were again similar between replicates, but details (*e.g.* tertiary interactions) were not (S10 Fig). The ion positions also differed between replicates, suggesting that the simulations were too short to equilibrate the ions fully, limiting the conclusions that can be drawn from simulations with $Mg^{2+}$.

Looking next at the results for the Vs-CoV-1 pseudoknot (cluster 2, beta-CoV), they were broadly similar to the results for the GLGC2 pseudoknot: this pseudoknot, too, shared a 3-stem architecture, with S3 appearing as an extended helix off the end of a 2-stem H-type pseudoknot (Fig 4). As above, L3 was longer than in the SARS-CoV-2 pseudoknot, and S1 and

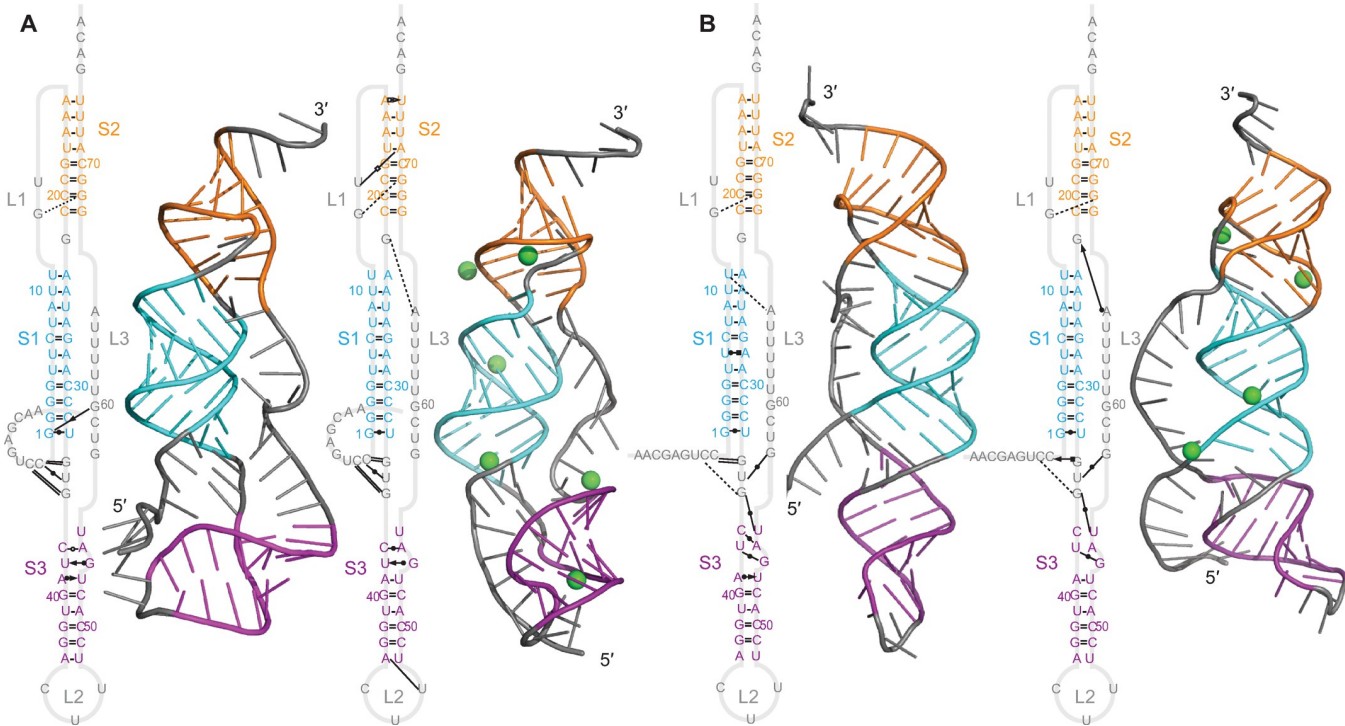

**Fig 4. Structural models for Vs-CoV-1 pseudoknot.** (A) Models with 5′ end threaded between S1 and S3, with (right) and without (left) $Mg^{2+}$ (ions shown in green). 3D structure representative of most populated cluster. (B) Models with unthreaded 5′ end.

S2 were fully base-paired, without any bulges. However, S3 contained bulged bases in the unthreaded model, S2 was notably longer in this pseudoknot compared to the others, and S2 also featured fewer tertiary contacts. The base-pairing was unchanged after MD simulations compared to the 2D predictions (S2A Fig). $Mg^{2+}$ had little effect on the number of tertiary contacts for either the threaded (Fig 4A) or unthreaded (Fig 4B) conformation. The fluctuations were again highest in L2, and lower in L3 despite its length (S11 Fig), as above.

## Alpha-CoV pseudoknots

Turning to the Anlong-44 pseudoknot (cluster 3, alpha-CoV), we found that it was distinctive from the three beta-CoV pseudoknots discussed so far in containing a fourth helix: here, the loop between S2 and S3 was long enough that an extra helix formed within it (Fig 5). Such a 4-stem architecture has not been reported previously for any frameshift-stimulatory pseudoknot. S1–S3 were roughly collinear in most replicates but generally unstacked (with the exception of S1/S3 in the unthreaded conformer), similar to the beta-CoV pseudoknots, but S4 protruded sideways out of the S1–S3 axis, changing the envelope of the RNA structure. L1 was the shortest of all the pseudoknots, at only 0–1 nt. As with the GLGC2 and Vs-CoV-1 pseudoknots, S1 and S2 were fully paired, but S3 contained one or more bulges, unlike the GLGC2 pseudoknot. The base-pairing changed slightly in some of the MD simulations compared to the 2D predictions (S2A Fig), shortening S4 and/or S3 by 1 bp. The network of tertiary interactions was much sparser than for the beta-CoVs, especially in S2, which included no tertiary

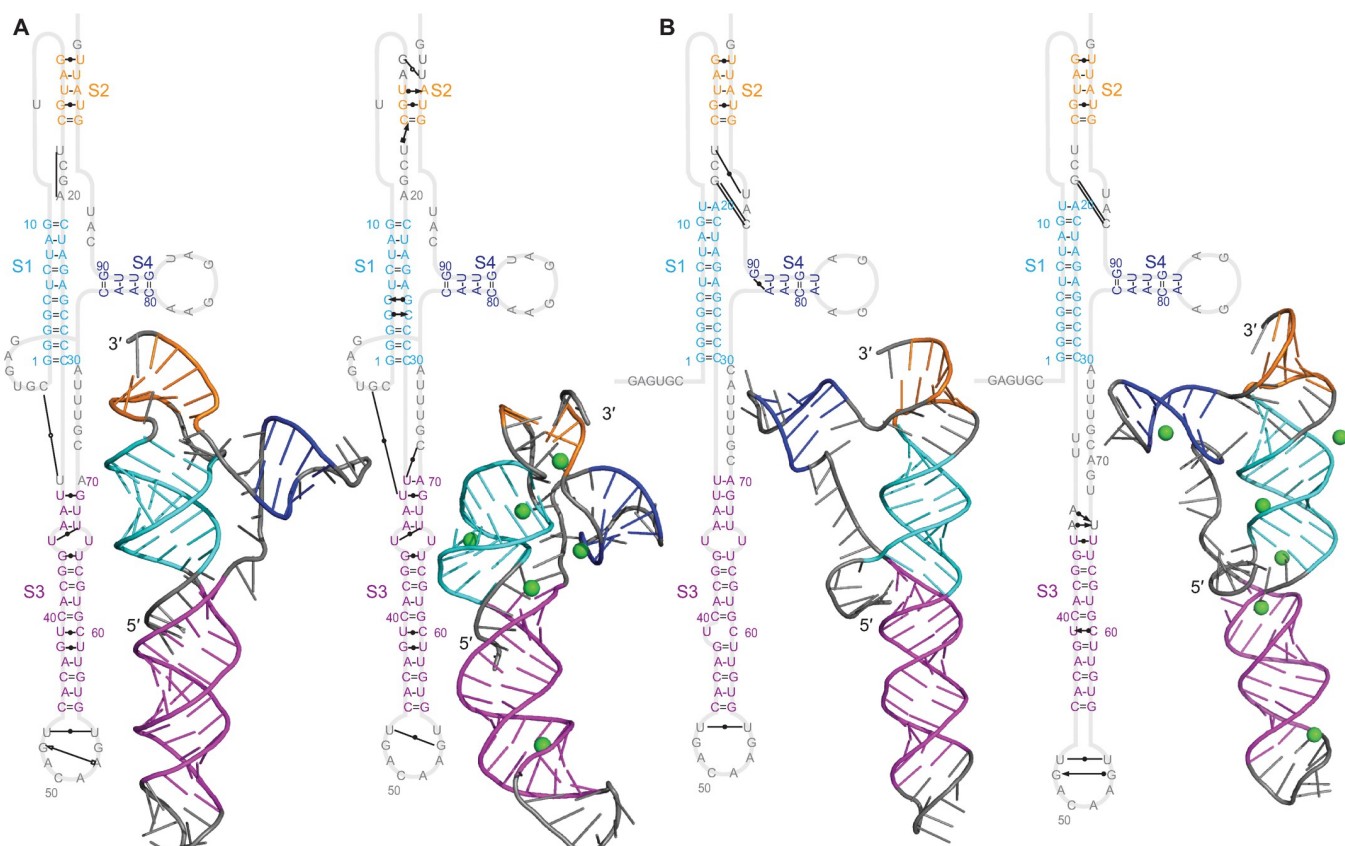

**Fig 5. Structural models for Anlong-44 pseudoknot.** (A) Models with 5′ end threaded between S1 and S3, with (right) and without (left) $Mg^{2+}$ (ions shown in green). 3D structure representative of most populated cluster. (B) Models with unthreaded 5′ end.

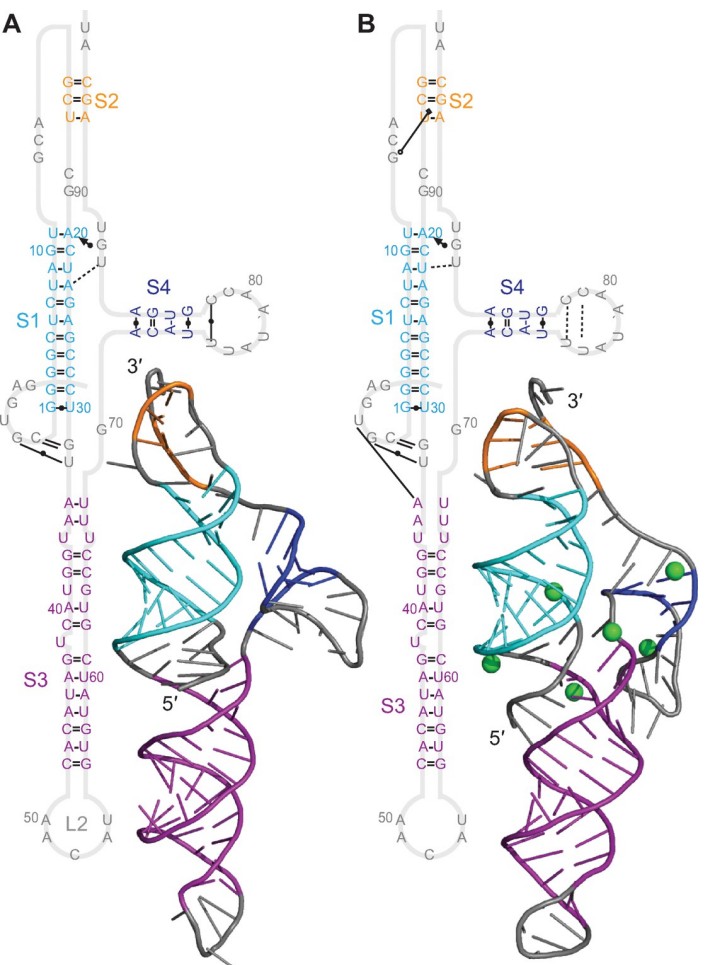

**Fig 6. Structural models for 21164–6 pseudoknot.** Models with 5′ end threaded between S1 and S3, (A) without $Mg^2$ $^+$ and (B) with $Mg^{2+}$ (ions shown in green). Secondary structures shown with tertiary interactions notated as in Ref. 38. 3D structure representative of most populated cluster.

contacts with L1 (possibly because the latter was so short). $Mg^{2+}$ ions had only a modest effect on the number of tertiary contacts for either the threaded (Fig 5A) or unthreaded (Fig 5B) conformation, although they disrupted some base-pairs at the top of S2 in the threaded structure and the top of S3 in the unthreaded structure, and they closed the S3 bulge and disrupted S1-S3 stacking in the unthreaded conformer. As with the other pseudoknots, fluctuations were highest in L2 (S12 Fig); here they were also generally higher in S4 than within any other stems, suggesting that the fourth stem is more dynamic than the others.

The 21164–6 pseudoknot (cluster 4, alpha-CoV) was predicted to have 4 stems, with S1–S3 arranged collinearly and S4 pointing off axis, similar to the Anlong-44 pseudoknot (Fig 6). Unlike the pseudoknots discussed above, however, no unthreaded conformer was predicted here. As with the Anlong-44 pseudoknot, S1 and S2 were fully paired whereas S3 contained at least one bulge, and the network of tertiary interactions was again much sparser than for the beta-CoVs. S2 was the shortest among all the pseudoknots, at only 3 base-pairs long. The base-pairing in S3 and S4 changed from the 2D predictions (S2A Fig), with S4 gaining 1 bp and S3 losing 1. $Mg^{2+}$ ions again had only a modest effect on the tertiary contact network, but they disrupted base-pairs at the top of S3 and the stacking of S1/S3. This disruption occurred in

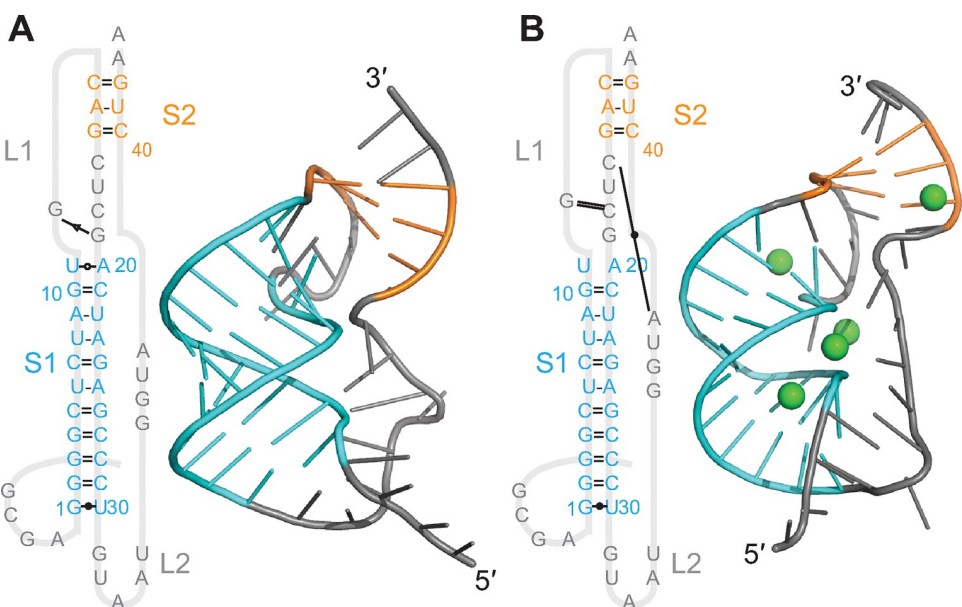

**Fig 7. Structural models for Anlong-8 pseudoknot.** Models with 5′ end threaded between S1 and S2, (A) without $Mg^{2+}$ and (B) with $Mg^{2+}$ (ions shown in green). 3D structure representative of most populated cluster.

replicate trajectories with $Mg^{2+}$, but only if an ion was present at the junction—stacking was preserved in the absence of $Mg^{2+}$ at the junction (S13 Fig). Fluctuations were largest in L2 and only modest in S4 (S14 Fig), indicating that S4 is less dynamic than for the Anlong-44 pseudo-knot, presumably owing to a shorter loop linking S3 and S4. Interestingly, the second-lowest energy prediction from FARFAR2 for this pseudoknot featured a qualitatively different fold, with S3 and S4 collinear and docked alongside S1 and S2 (S3 Fig), but it was not stable in MD simulations: in 3 replicate trajectories, S3 undocked after ~350 ns and reoriented to become collinear with S1 as in the other predictions.

We next considered the Anlong-8 pseudoknot (cluster 5, alpha-CoV). As with the Anlong-44 and 21164–6 alpha-CoV pseudoknots, this one also differed from the 3-stem architecture of the pseudoknots from the beta-CoV clusters. However, it had one stem fewer rather than one more: the loop between S1 and S2 was sufficiently short that no S3 could be formed (Fig 7). Despite looking more like a standard H-type pseudoknot, however, it still featured the 5′ end threading encountered in all the other bat-CoV pseudoknots, and indeed no unthreaded struc-ture was predicted (as for 21164–6). S1 and S2 were fully paired, and the network of tertiary interactions was very sparse, lacking any tertiary contacts with S2. Base-pairing after MD sim-ulations was unchanged compared to 2D predictions (S2A Fig). Fluctuations were relatively small even in the loops (S15 Fig), likely owing to this pseudoknot having the most compact structure of the set studied here.

Unlike the other alpha-CoV pseudoknots described above, the Rs4236 pseudoknot (cluster 6, alpha-CoV) had the same number of stems as the beta-CoV pseudoknots (three), but with S3 oriented differently: it was off-axis from S1, instead of collinear (Fig 8). This off-axis stem was larger than those in any of the other pseudoknots featuring off-axis stems (7 bp compared to 4–5 bp), leading to an exaggerated T-shaped envelope. L2 at the end of the S3 protrusion was also the locus of the largest structural fluctuations (S16 Fig). Base-pairing in S1 and S3 was unchanged after MD simulations compared to the 2D predictions (S2A Fig), but S2 was

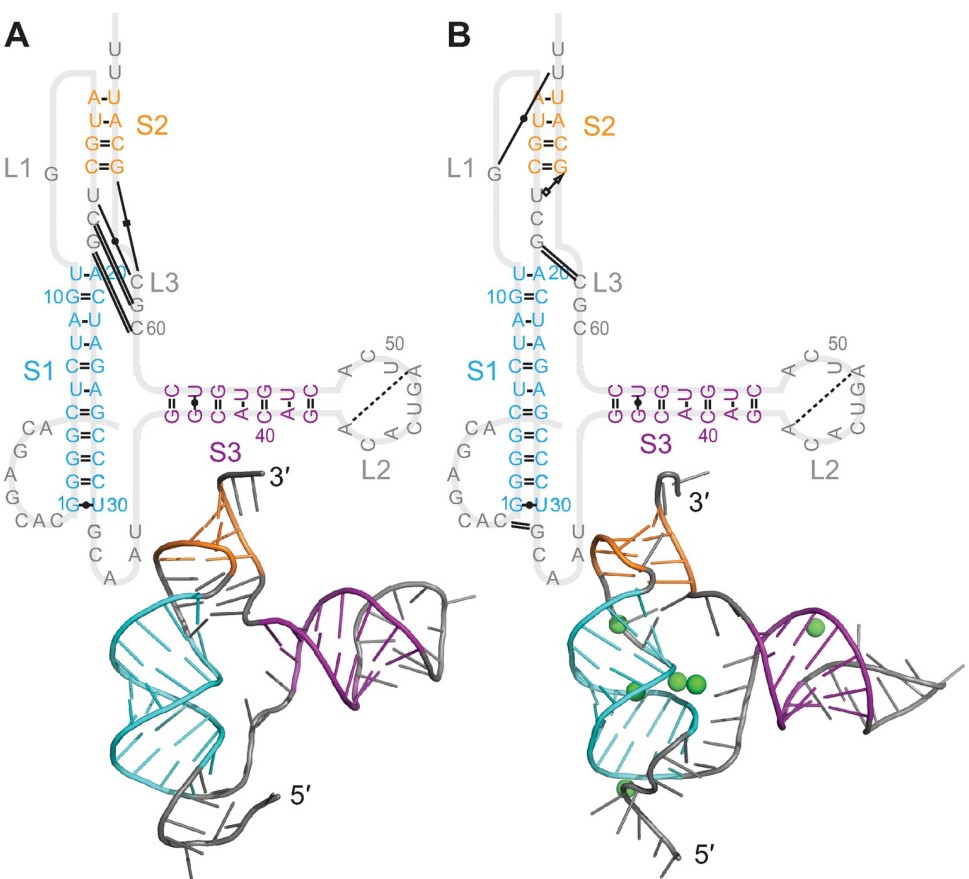

**Fig 8. Structural models for Rs4236 pseudoknot.** Models with 5′ end threaded between S1 and S2, (A) without $Mg^{2+}$ and (B) with $Mg^{2+}$ (ions shown in green). 3D structure representative of most populated cluster.

extended by additional base-pairs (such as those seen in Fig 8A) in some but not all of the replicates starting from different 3D predictions.

Turning to the Anlong-172 pseudoknot (cluster 7, alpha-CoV), it was most similar to the Rs4236 pseudoknot: like the latter, it featured 3 stems, with S3 oriented off-axis (Fig 9). All stems were fully paired, without bulges, but none were stacked. Tertiary interactions were sparse, but increased with $Mg^{2+}$ present, particularly in S2 and S3. The structures containing $Mg^{2+}$ were seen to be more compact compared to those without Mg across multiple simulations (Figs 9 and S10C). Fluctuations were largest in L2, as in all the 3-stem pseudoknots (S17 Fig).

Finally, we considered the Neixiang-64 pseudoknot (cluster 8, alpha-CoV). Although the consensus 2D structure prediction featured three stems (S2A Fig), the predicted S3 was short, and we found that it dissociated in MD simulations before ~500 ns (within the equilibration phase) in multiple replicates, implying that it was not stable. This 3-stem structure was in fact an outlier among the base-pairing predictions for cluster 8: the other pseudoknots in this cluster were predicted to have only 2 stems (S1 Table). We therefore turned to the lowest-energy prediction that was not part of the initial consensus, a 2-stem pseudoknot predicted by pKiss (S2B Fig) that was similar to the structures predicted for the rest of cluster 8. This structure did indeed remain stable in the MD simulations, with the only change to the base-pairing from the 2D prediction being that 1 nt from L1 contributed to a 1-bp extension of S2. As with the Anlong-8 pseudoknot, therefore, this pseudoknot differed from the three-stem architecture of

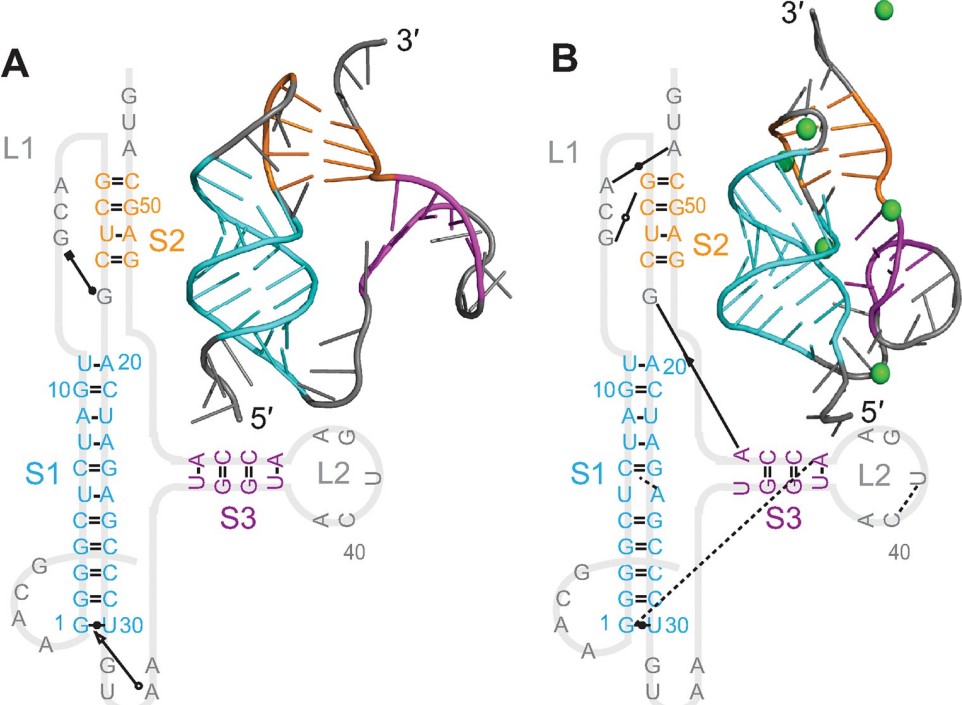

**Fig 9. Structural models for Anlong-172 pseudoknot.** Models with 5′ end threaded between S1 and S2, (A) without Mg²⁺ and (B) with Mg²⁺ (ions shown in green). 3D structure representative of most populated cluster.

the pseudoknots from the beta-CoV clusters, having one less stem such that S3 was replaced with a longer L2 (Fig 10). As a result, the Neixiang-64 pseudoknot was the second-most compact of the representative structures studied here. Unlike the Anlong-8 pseudoknot, however, the Neixiang-64 pseudoknot featured both threaded and unthreaded conformers. Interestingly, although S1 and S2 were closely stacked, tertiary interactions were much sparser than in the beta-CoVs. Mg²⁺ ions showed little effect on the threaded conformer (Fig 10A), but

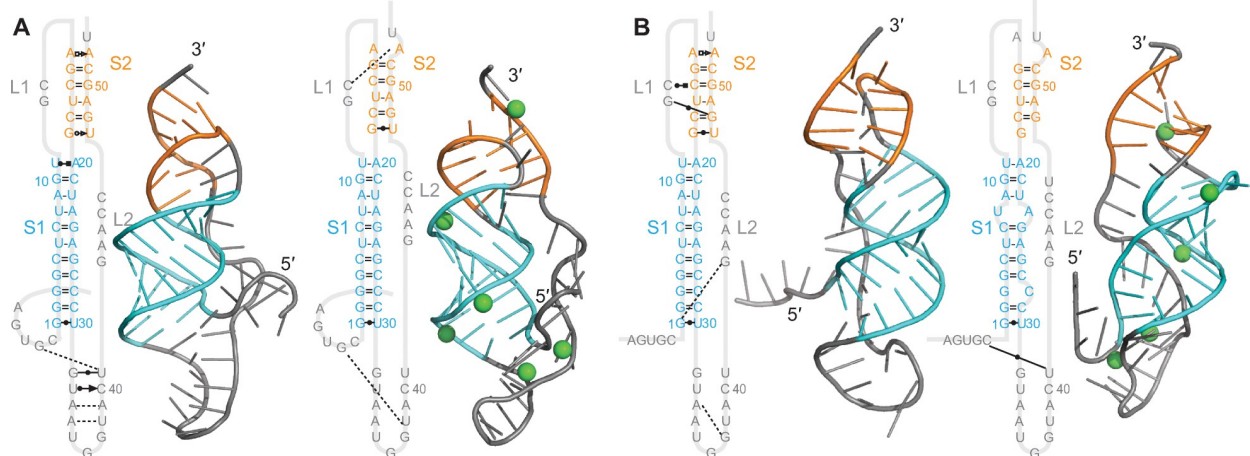

**Fig 10. Structural models for Neixiang-64 pseudoknot.** (A) Models with 5′ end threaded through L2, with (right) and without (left) Mg²⁺ (ions shown in green). 3D structure representative of most populated cluster. (B) Models with unthreaded 5′ end.

induced a distortion in the unthreaded conformer that shortened S2 and opened a bulge in S1 (Fig 10B). The fluctuations were once again largest in L2 (S18 Fig).

## Discussion

Aside from extensive studies of the SARS-CoV-2 pseudoknot, relatively little structural information exists for frameshift-stimulatory pseudoknots from CoVs. The work here extends our understanding of the structures that may be formed by CoV pseudoknots considerably, through all-atom simulations of eight different CoV pseudoknots. Notably, these pseudoknots are representative of the different types of pseudoknots found in bat CoVs that are distinct from SARS-like CoVs. Together with previously published models of the SARS-CoV-2 pseudoknot structure [18–23], the models presented here provide an overview of the range of structures to be expected in bat CoVs—the ultimate source of past human CoV disease outbreaks and the likely source of any new CoV zoonoses in coming decades.

The most notable feature of these results is that all of the pseudoknots demonstrated the ability to form an unusual fold topology in which the 5′ end is threaded through a ring formed by the closure of the pseudoknotted base-pairs in S2. In half of the pseudoknots, a second fold topology was also possible, in which the 5′ end was unthreaded along the outside of this ring. This same behavior—threaded and unthreaded fold topologies—was also seen for the SARS-CoV-2 pseudoknot [18,20–23,26]. Threaded 5′ ends had never previously been reported in any frameshift-stimulatory pseudoknot, and had been seen only in viral exoribonuclease-resistant RNAs (xrRNAs) [24,45], which like the CoV pseudoknots can also form unthreaded conformers [25]. Our results indicate that 5′-end threading in such a 'ring-knot' topology is likely a feature common to all CoV frameshift-stimulatory pseudoknots, one that appears so far to be unique to CoVs, distinguishing them from other viral pseudoknots. Intriguingly, although the two previous examples of RNA structures with ring-knots (SARS-CoV-2 pseudoknot and Zika virus xrRNA) both involved threading through a three-helix junction, the pseudoknots from Anlong-8 and Neixiang-64 presented here contained only two stems, implying that the third stem is not, in fact, essential to forming the ring-knot, as long as the loop forming the ring is sufficiently large. Given that the threaded conformation is the dominant one seen in structural and dynamic studies of the SARS-CoV-2 pseudoknot [18,20–23,26], the same is likely true for all the bat-CoV pseudoknots studied here. However, the unthreaded conformers may still be important functionally, as they increase the conformational heterogeneity of the frameshift signal and he nce likely contribute to the relatively high levels of −1 PRF stimulated by these pseudoknots [46,47].

Although the presence of 5′-end threading was universal in these pseudoknots, the number of stems was not. Whereas the pseudoknots representative of the three beta-CoV clusters (GLGC2, Vs-CoV-1, and SARS-CoV-2) all contained three stems, as did those for two of the alpha-CoV clusters (Rs4236 and Anlong-172), those representative of the other four alpha-CoV clusters did not: the pseudoknots from Anlong-44 and 21164–6 contained four stems, whereas those from Anlong-8 and Neixiang-64 contained only two. These results reflect a greater structural diversity among the alpha-CoVs (implicit already in the larger number of alpha-CoV clusters seen in the phylogenetic tree). Furthermore, this increased diversity extends to the 3D arrangement of the helices: whereas all the beta-CoV pseudoknots had globally similar folds, with all three stems arranged roughly collinearly, the four alpha-CoV pseudoknots with more than 2 stems (Rs4236, Anlong-172, 21164–6, and Anlong-44) featured one stem perpendicular to the long axis formed by the other stems, so that even the 3-stem alpha-CoV pseudoknots were structurally distinct from the beta-CoV pseudoknots. Ultimately, the large-scale differences in structural architecture among the bat-CoV pseudoknots derived primarily from variations in the length of the segment looping between S1 and S2. Whereas S1

was of similar length for all pseudoknots, at 10–11 bp, and S2 varied somewhat, from 3–8 bp, the segment between them ranged very widely, from 9–63 nt—too short at one extreme to allow a third helix, but long enough at the other to accommodate one or even two extra helixes in various orientations.

These differences in structural composition raise the question of whether the number of stems and/or their orientation might be related to differences in −1 PRF stimulation efficiency among CoV frameshift signals that were reported previously. The GLGC2, Vs-CoV-1, Neixiang-64, and SARS-CoV-2 pseudoknots, featuring 2 or 3 helices aligned on a single long axis, all induced −1 PRF at relatively high levels of ~25–35% in rabbit reticulocyte lysate, whereas the Anlong-44 pseudoknot, featuring 4 helices with one orthogonal to the long axis, only led to −1 PRF ~10% of the time. [17]. The lack of S3 does not seem to affect the ability of the Neixiang-64 pseudoknot to stimulate −1 PRF at close to the same level as the pseudoknots from GLGC2, Vs-CoV-1, and SARS-CoV-2 [17]. Indeed, this result makes sense in light of earlier work showing that deleting S3 from the SARS-CoV pseudoknot had little effect on −1 PRF efficiency [44]. We speculate that the low −1 PRF levels for the Anlong-44 pseudoknot might arise from its 4-stem architecture or the presence of an off-axis helix: S4 might reduce −1 PRF by interfering sterically with contacts between the pseudoknot and ribosome, or with the ability of S3 to bend with respect to S1/S2 in the presence of the ribosome as seen in cryo-EM imaging of the SARS-CoV-2 pseudoknot on the ribosome [18], although S3 does not appear to be more rigid than in the other pseudoknots in the simulations (S19 Fig). Alternatively, the low −1 PRF efficiency might arise from this pseudoknot having a less stable S2 containing fewer G:C pairs than in the other pseudoknots whose frameshifting has been measured, given that disrupting S2 in the SARS-CoV and SARS-CoV-2 pseudoknots significantly reduces −1 PRF levels [44,48]. Future measurements of −1 PRF for bat-CoV pseudoknots that have not yet been studied (*e.g.* the 4-stem 21164–6 versus the off-axis 3-stem Rs4236 and Anlong-172) may help clarify the relative importance of number of stems and off-axis stem orientation for influencing −1 PRF levels.

A second feature that is very similar across the panel of bat-CoV pseudoknots is stem 1. S1 had a very similar length in all the representative pseudoknots studied, 10 or 11 base-pairs, and a high level sequence of sequence identity: 70–100% of base-pairings were the same in pairwise comparisons (S2 Fig). As a result, the conformation of S1 in the 5′-threaded conformers was effectively the same for all the pseudoknots (Fig 11, cyan); these conformations also matched those of S1 in the SARS-CoV-2 pseudoknot (Fig 11, black). Most importantly, this constancy in the structure of S1 positioned the 5′ end of the pseudoknot (Fig 11, red) for ring-knot threading in the same way for each of the pseudoknots, regardless of the number of stems in the pseudoknot. The loops defining the ring through which the 5′ end is threaded were more variable (Fig 11, grey), but the ring region enclosing the threaded end nevertheless shared broadly similar features in all the pseudoknots.

These structural similarities between pseudoknots representative of the different varieties found in bat CoVs suggest that it should be possible to find small-molecule ligands that can inhibit −1 PRF in a broad spectrum of CoVs, and indeed some such ligands have been found [9,17]. The pseudoknots from the 3 beta-CoV clusters appear most similar structurally, suggesting that it may be easiest, if searching for −1 PRF inhibitors that can act against some subset of CoVs, to do so by targeting beta-CoVs specifically, finding ligands that bind the pseudoknots. Supporting this view, the compound merafloxacin was found to be far more active at inhibiting −1 PRF in beta-CoVs than in alpha CoVs, doing so for five human beta-CoVs and one bat beta-CoV but no human alpha-CoVs and only one bat alpha-CoV [9,17]. However, other −1 PRF inhibitors were found to be active against frameshift signals from both some beta-CoVs and some alpha-CoVs, suggesting it is possible to find inhibitors active across both CoV families. Indeed, the compound nafamostat was found to inhibit −1 PRF to some

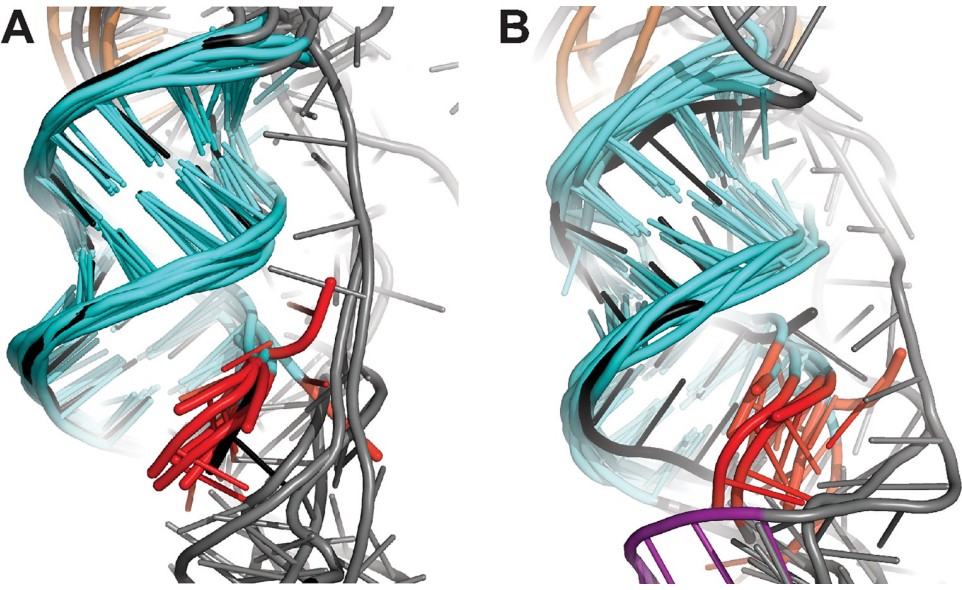

**Fig 11. Structural similarities in S1 and the ring-knot region.** Alignment of S1 in pseudoknots from 8 bat CoVs (cyan) and SARS-CoV-2 (black) for models with 5′-end threading, (A) without and (B) with $Mg^{2+}$. 5′ end shown in red.

extent for all five clusters of bat-CoV frameshift signals [17], highlighting the potential for ligands to act as broad-spectrum anti-CoV drugs by targeting −1 PRF.

As a final note, the structural models outlined above should be helpful for future studies of bat-CoV pseudoknots, as experimental characterizations of CoVs are extended from SARS-CoV-2 to the bat CoVs that are likely sources of future zoonotic CoV diseases. These models can be used to help interpret small- and wide-angle solution x-ray scattering measurements, by comparing the scattering profiles predicted from model structures to those observed empirically [49,50], as well as single-molecule measurements of folding dynamics, which can detect the formation of distinct sub-populations with different conformations and verify the presence of alternative threading of the 5′ end [26]. By identifying features that are common across a wide range of CoV pseudoknots, these models may also support efforts to find broad-spectrum anti-CoV therapeutics that act by inhibiting −1 PRF.

## Supporting information

**S1 Data. Multiple sequence alignment of bat-CoV frameshift signals.** PDF file containing multiple sequence alignment used for clustering bat-CoV pseudoknots.
(PDF)

**S2 Data. Sequences and predicted secondary structures.** Excel file containing consensus 2D structure predictions from S1 Table aligned with pseudoknot sequences.
(XLSX)

**S3 Data. (Model_PDB_files.zip): structures from computational models.** PDB files for the top 3 representative clusters from each conformer (threaded/unthreaded) of each pseudo-knot described in the main text, along with PDB files for the representative clusters shown in S8–S18 Figs.
(ZIP)

**S1 Table. Secondary structure predictions.** Consensus 2D structure predictions for all pseudoknots in Fig 1, in dot-bracket notation. Source viruses listed by NCBI Virus accession number. Pseudoknots selected as representative examples from each cluster for structural characterization colored in red. Predictors contributing to consensus shown by tick mark. Note that the consensus prediction for SARS-CoV-2 (accession number NC_045512) matches the secondary restructure reported in cryo-EM structures of the pseudoknot [18,21].
(DOC)

**S1 Fig. R-scape analysis of bat-CoV multiple sequence alignment for conservation of SARS-CoV-2 pseudoknot secondary structure.** Stem 1 is highly conserved across the multiple sequence alignment, but stem 3 less so, consistent with the large variability in S3 in the predicted structures. R-scape does not evaluate stem 2 because the pseudoknotted base pairs are ignored by Infernal when constructing the covariance model. The extra stem is a hairpin downstream of the pseudoknot in the SARS-CoV-2 genome that is also less conserved.
(TIF)

**S2 Fig. Base-pairing predictions for bat-CoV pseudoknots.** (A) Lowest-energy consensus predictions from pKiss, Hotknots, PKNOTS, and NUPACK. Cyan: S1; gold: S2; purple: S3; blue: S4; green: slippery sequence. (B) Alternative prediction for Neixiang-64 from pKiss.
(TIF)

**S3 Fig. Initial predictions for 5′-threaded 3D structures.** Predicted structures that featured 5′ end threading for each of the eight representative bat-CoV pseudoknots. Structures illustrated in order of increasing energy from left to right. Lowest-energy structures selected for use in MD simulations shown in green boxes. Next-lowest energy threaded conformers selected for MD replicates shaded in brown.
(TIF)

**S4 Fig. Initial predictions for other 3D structures.** (A) Predicted structures without 5′ end threading for bat-CoV pseudoknots featuring unthreaded conformers. (B) Predicted structures rejected because they contained topological knots or 3′ end threading. (C) Alternative predictions for Neixiang-64 featuring 2-stem architecture. Structures illustrated in order of increasing energy from left to right. Lowest-energy structures for each conformer selected for use in MD simulations shown in green boxes. Next-lowest energy threaded conformers selected for MD replicates shaded in brown.
(TIF)

**S5 Fig. Initial Mg$^{2+}$ ion positions in MD simulations.** Mg$^{2+}$ ions positioned by hand using MOE.
(TIF)

**S6 Fig. Two-dimensional RMSD comparisons of replicate trajectories.** Pairwise RMSD for the structured regions of replicate trajectories for the (A) GLGC2, (B) Vs-CoV-1, (C) Anlong-44, (D) 21164–6, (E) Anlong-8, (F) Rs4236, (G) Anlong-172, and (H) Neixiang-64 pseudoknots.
(TIF)

**S7 Fig. SARS-CoV-2 pseudoknot structures.** (A) Structures obtained from cryo-EM images of the pseudoknot on the ribosome (left, PDB ID: 7O7Z) and in isolation (right, PDB ID: 6XRZ). (B) Structures obtained from x-ray crystallography, complexed with a chaperone (left, PDB ID: 7MLX) and in isolation (middle, PDB ID: 7LYJ; right, PDB ID: 7MKY). (C) Threaded conformer from MD simulation of lowest-energy prediction by FARFAR2 [20]. (D)

Experimentally observed base-pairing [18,21–23,48]. Asterisked base-pairs are not consistent between structures. Note that the computational model, obtained from blind 3D structure prediction as in the procedure used for the bat-CoV pseudoknots, is qualitatively similar to the experimental structures.
(TIF)

**S8 Fig. MD simulations results for GLGC2 pseudoknot.** (A) Overlay of the 3D structures of the three most populated clusters from simulations of the 5′-threaded structure in Fig 3 without Mg²⁺. Cyan: S1, gold: S2, purple: S3. Top inset: RMSD vs time. Portions of this trajectory in which the top three clusters are occupied are indicated in color (cyan: cluster 1, orange: cluster 2, green: cluster 3). Bottom inset: RMSF for each residue. (B) The same for the 5′-threaded structure in Fig 3 with Mg²⁺ (ions not shown for clarity). (C) The same for the unthreaded structure in Fig 3 without Mg²⁺. (D) The same for the unthreaded structure in Fig 3 with Mg²⁺ (ions not shown for clarity). (E) The top cluster from replicate simulations starting from the 3 lowest-energy FARFAR2 predictions (lowest-energy on left), showing qualitatively similar results. Right: average RMSF over the replicates; error bars indicate s.e.m.
(TIF)

**S9 Fig. Replication of MD simulation results in different trajectories from same starting structure.** (A) Results for the most-occupied cluster of GLGC2 pseudoknot are qualitatively the same from 3 separate replicates. (B) Same for Vs-CoV-1 pseudoknot.
(TIF)

**S10 Fig. Simulations with different initial Mg²⁺ ion positions.** (A) Secondary structure, tertiary contacts, and 3D structure of the most-occupied cluster from MD simulations of the threaded conformer of the GLGC2 pseudoknot with Mg²⁺ ions initially placed at three different sets of positions (i–iii). Top: initial ion positions in predicted 3D structure; bottom: result after MD simulation. (B) 3D structures and ion positions for top 3 clusters for each of the 3 initial ion positions, showing ion positions are not fully equilibrated. (C) Secondary structure, tertiary contacts, and 3D structure of the most-occupied cluster from MD simulations of the threaded conformer of the Anlong-172 pseudoknot with Mg²⁺ ions initially placed at three different sets of positions (i–iii).
(TIF)

**S11 Fig. MD simulations results for Vs-CoV-1 pseudoknot.** (A) Overlay of the 3D structures of the three most populated clusters from simulations of the 5′-threaded structure in Fig 4 without Mg²⁺. Cyan: S1, gold: S2, purple: S3. Top inset: RMSD vs time. Portions of this trajectory in which the top three clusters are occupied are indicated in color (cyan: cluster 1, orange: cluster 2, green: cluster 3). Bottom inset: RMSF for each residue. (B) The same for the 5′-threaded structure in Fig 4 with Mg²⁺ (ions not shown for clarity). (C) The same for the unthreaded structure in Fig 4 Mg²⁺. (D) The same for the unthreaded structure in Fig 4 with Mg²⁺ (ions not shown for clarity).
(TIF)

**S12 Fig. MD simulations results for Anlong-44 pseudoknot.** (A) Overlay of the 3D structures of the three most populated clusters from simulations of the 5′-threaded structure in Fig 5 without Mg²⁺. Cyan: S1, gold: S2, purple: S3, blue: S4. Top inset: RMSD vs time. Portions of this trajectory in which the top three clusters are occupied are indicated in color (cyan: cluster 1, orange: cluster 2, green: cluster 3). Bottom inset: RMSF for each residue. (B) The same for the 5′-threaded structure in Fig 5 with Mg²⁺ (ions not shown for clarity). (C) The same for the unthreaded structure in Fig 5 without Mg²⁺. (D) The same for the unthreaded structure in

Fig 5 with $Mg^{2+}$ (ions not shown for clarity). (E) Top cluster from replicate simulations starting from 3 lowest-energy FARFAR2 predictions (lowest-energy on top), showing qualitatively similar results with the exception of the orientation of S3. Right: average RMSF over 3 replicates; error bars indicate s.e.m.
(TIF)

**S13 Fig. $Mg^{2+}$ affects S1/S3 stacking in 21164–6 pseudoknot.** Replicates with $Mg^{2+}$ ions at different initial positions show that S1/S3 stacking is disrupted if $Mg^{2+}$ is bound at the S1/S3 interface (A), but preserved otherwise (B).
(TIF)

**S14 Fig. MD simulations results for 21164–6 pseudoknot.** (A) Overlay of the 3D structures of the three most populated clusters from simulations of the 5′-threaded structure in Fig 6 without $Mg^{2+}$. Cyan: S1, gold: S2, purple: S3, blue: S4. Top inset: RMSD vs time. Portions of this trajectory in which the top three clusters are occupied are indicated in color (cyan: cluster 1, orange: cluster 2, green: cluster 3). Bottom inset: RMSF for each residue. (B) The same for the 5′-threaded structure in Fig 6 with $Mg^{2+}$ (ions not shown for clarity). (C) Top cluster from replicate simulations starting from lowest- and 3rd-lowest-energy FARFAR2 predictions (lowest-energy on top), showing qualitatively similar results; 2nd-lowest energy structure had a different fold that was not stable in MD simulation. Right: average RMSF over replicates.
(TIF)

**S15 Fig. MD simulations results for Anlong-8 pseudoknot.** (A) Overlay of the 3D structures of the three most populated clusters from simulations of the 5′-threaded structure in Fig 7 without $Mg^{2+}$. Cyan: S1, gold: S2, purple: S3, blue: S4. Top inset: RMSD vs time. Portions of this trajectory in which the top three clusters are occupied are indicated in color (cyan: cluster 1, orange: cluster 2, green: cluster 3). Bottom inset: RMSF for each residue. (B) The same for the 5′-threaded structure in Fig 7 with $Mg^{2+}$ (ions not shown for clarity). (C) Top cluster from replicate simulations starting from 3 lowest-energy FARFAR2 predictions (lowest-energy on left), showing qualitatively similar results. Right: average RMSF over 3 replicates; error bars indicate s.e.m.
(TIF)

**S16 Fig. MD simulations results for Rs4236 pseudoknot.** (A) Overlay of the 3D structures of the three most populated clusters from simulations of the 5′-threaded structure in Fig 8 without $Mg^{2+}$. Cyan: S1, gold: S2, purple: S3, blue: S4. Top inset: RMSD vs time. Portions of this trajectory in which the top three clusters are occupied are indicated in color (cyan: cluster 1, orange: cluster 2, green: cluster 3). Bottom inset: RMSF for each residue. (B) The same for the 5′-threaded structure in Fig 8 with $Mg^{2+}$ (ions not shown for clarity). (C) Top cluster from replicate simulations starting from 3 lowest-energy FARFAR2 predictions (lowest-energy on left), showing qualitatively similar results, with the exception of S3 orientation. Right: average RMSF over 3 replicates; error bars indicate s.e.m.
(TIF)

**S17 Fig. MD simulations results for Anlong-172 pseudoknot.** (A) Overlay of the 3D structures of the three most populated clusters from simulations of the 5′-threaded structure in Fig 9 without $Mg^{2+}$. Cyan: S1, gold: S2, purple: S3, blue: S4. Top inset: RMSD vs time. Portions of this trajectory in which the top three clusters are occupied are indicated in color (cyan: cluster 1, orange: cluster 2, green: cluster 3). Bottom inset: RMSF for each residue. (B) The same for the 5′-threaded structure in Fig 9 with $Mg^{2+}$ (ions not shown for clarity). (C) Top cluster from replicate simulations starting from 3 lowest-energy FARFAR2 predictions (lowest-energy on

left), showing qualitatively similar results. Right: average RMSF over 3 replicates; error bars indicate s.e.m.
(TIF)

**S18 Fig. MD simulations results for Neixiang-64 pseudoknot.** (A) Overlay of the 3D structures of the three most populated clusters from simulations of the 5′-threaded structure in Fig 10 without $Mg^{2+}$. Cyan: S1, gold: S2. Top inset: RMSD vs time. Portions of this trajectory in which the top three clusters are occupied are indicated in color (cyan: cluster 1, orange: cluster 2, green: cluster 3). Bottom inset: RMSF for each residue. (B) The same for the 5′-threaded structure in Fig 10 with $Mg^{2+}$ (ions not shown for clarity). (C) The same for the unthreaded structure in Fig 10 without $Mg^{2+}$. (D) The same for the unthreaded structure in Fig 10 with $Mg^{2+}$ (ions not shown for clarity). (E) Top cluster from replicate simulations starting from 3 lowest-energy FARFAR2 predictions (lowest-energy on left), showing qualitatively similar results, with the exception of L2. Right: average RMSF over 3 replicates; error bars indicate s.e.m.
(TIF)

**S19 Fig. Angular fluctuations of S3 with respect to S1.** The S1-S3 angle was measured from 5,000 frames (one every 0.1 ns) of the threaded conformers without $Mg^{2+}$ and the standard deviation calculated as a measure of the flexibility of S3, for the 3- and 4-stem pseudoknots whose frameshifting has been measured experimentally.
(TIF)

## Acknowledgments

We thank Compute Canada for providing access to computational resources for this project. We thank Meng Zhao for help with tertiary interaction notation.

## Author Contributions

**Conceptualization:** Michael T. Woodside.

**Data curation:** Rohith Vedhthaanth Sekar.

**Formal analysis:** Rohith Vedhthaanth Sekar.

**Funding acquisition:** Michael T. Woodside.

**Investigation:** Rohith Vedhthaanth Sekar, Patricia J. Oliva.

**Methodology:** Rohith Vedhthaanth Sekar, Patricia J. Oliva.

**Project administration:** Michael T. Woodside.

**Supervision:** Michael T. Woodside.

**Visualization:** Rohith Vedhthaanth Sekar, Michael T. Woodside.

**Writing – original draft:** Rohith Vedhthaanth Sekar, Michael T. Woodside.

**Writing – review & editing:** Rohith Vedhthaanth Sekar, Patricia J. Oliva, Michael T. Woodside.

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
