## [Decision Letter · Decision Letter 0]

14 Sep 2022

Dear Dr. Woodside,

Thank you very much for submitting your manuscript "Modelling the structures of frameshift-stimulatory pseudoknots from representative bat coronaviruses" for consideration at PLOS Computational Biology.

As with all papers reviewed by the journal, your manuscript was reviewed by members of the editorial board and by several independent reviewers. In light of the reviews (below this email), we would like to invite the resubmission of a significantly-revised version that takes into account the reviewers' comments.

We apologize for the delay in decision. As some reviews came in, we wanted to obtain

more points of view to reach a balanced decision.

As you can see, the reviews vary, but the critiques share common points.

While interesting, there is a consensus that the work requires further structural validation and comparative sequence analysis to extract biological meaning from the results on different coronavirus families.

Several aspects of the methodology also require clarification, and improvement in the relevant section is requested.

While we appreciate that you are addressing a challenging problem, the work needs to go further to make the biological connections convincing.

Please address in full all comments of the reviewers.

We cannot make any decision about publication until we have seen the revised manuscript and your response to the reviewers' comments. Your revised manuscript is also likely to be sent to reviewers for further evaluation.

Sincerely,

Tamar Schlick

Academic Editor

PLOS Computational Biology

Nir Ben-Tal

Section Editor

PLOS Computational Biology

We apologize for the delay in decision. As some reviews came in, we wanted to obtain

more points of view to reach a balanced decision.

As you can see, the reviews vary, but the critiques share common points.

While interesting, there is a consensus that the work requires further structural validation and comparative sequence

analysis to extract biological meaning from the results on different coronavirus families.

Several aspects of the methodology also require clarification, and improvement in the relevant section is requested.

While we appreciate that you are addressing a challenging problem, the work needs to go further to make the biological

connections convincing.

Please address in full all comments of the reviewers.

Reviewer's Responses to Questions

**Comments to the Authors:**

Reviewer #1: In this study, the authors provide representative bat-CoVs frameshift-stimulatory pseudoknot structures that are distinct from SARS-like CoVs and study their key conformational features shedding light on frameshifting mechanisms and anti-CoV drugs. Two beta-CoV sequences and two alpha-CoV sequences are selected after clustering based on genetic distances. Pseudoknot structures are obtained from computational predictions of 2D and 3D structures in combination with microsecond molecular dynamics simulations. Although the four selected representative pseudoknots have different numbers of stems, all of them are predicted to adopt two folding topologies: 5’ end threaded through a ring at stem junction and unthreaded along the outside of the ring, same as SARS-CoV-2 frameshift pseudoknot. Stem 1 in the representative pseudoknots studied here has high similarities in both length and sequence identity, and S1 conformations also match those in SARS-CoV-2 frameshift pseudoknot. Based on these findings, the authors suggest the possibility to find broad-spectrum anti-CoV therapeutics by inhibiting -1 programmed ribosomal frameshifting.

The study is well-designed - from sequence to structure and function. The predicted structures are carefully selected - using consensus lowest-energy 2D structures and excluding disfavored 3D structures. The findings in this work support previous discoveries from SARS-CoV-2 pseudoknot in the 5’-end threaded ring-knot conformation and generalize the insights to a wide range of CoV pseudoknots. However, some of the unique findings in this work can be strengthened, such as different RNA foldings introduced by variations in sequence.

Major points：

The authors performed clustering on 48 bat-CoV sequences and selected representative sequences in this study. However, how sequence affects the RNA conformation is not clearly stated. The beta-CoV pseudoknots adopt a three-stem architecture while alpha-CoV pseudoknots don’t. Is this a coincidence? Does any of the sequences favor a long and stable pseudoknot Stem 2? A longer L3 is reported in several pseudoknots in this work compared with SARS-CoV-2. Is this due to sequence variations?

Minor points:

1. I was wondering if the authors tested and examined the Mg2+ binding sites or the convergence of MD trajectories regarding the initial Mg2+ ions placement. The results that the presence of Mg2+ increased the fluctuations of L3 in the two unthreaded beta-CoV pseudoknots and disrupted some base pairs in the two unthreaded alpha-CoV pseudoknots are very interesting.

2. The authors reported alternative 2D structures of KF294282, and some 3-stem 2D structures were not stable in MD simulations. Did these MD trajectories (starting from different predicted models of this pseudoknot) converge to the same 2-stem 2D structure?

Reviewer #2: The review is uploaded as an attachment.

Reviewer #3: In their manuscript "Modelling the structures of frameshift-stimulatory pseudoknots from representative bat coronaviruses", Sekar et al. employed established RNA structure prediction algorithms and subsequent all-atom MD simulations to model the structures of RNA pseudoknots crucial for the proliferation of coronaviruses. Knowledge of these structures might have important implications for finding antiviral drugs.

The manuscript tackles an important question and is very clearly written. I enjoyed reading it!

The authors miss several opportunities to validate their methods and results, which would strengthen their conclusions.

First, the SARS-CoV-2 pseudoknot was not considered because a wealth of structural information already exists. However, this structural information would render this pseudoknot a good candiate to test the structure prediction protocol by comparing the results with the available experimental data.

Second, for each considered pseudoknot and ion concentration, only one simulation was carried out and the simulations were considered equlibrated after 500 ns. How was it determined if a simulation is in equilibrium?

If the system is equilibrated, multiple simulations of the same length would give the same results and support the notion that equilibrium is reached.

Replicates of simulations are standard in the field to estimate the statistical errors and to prevent that the reported results are merely anecdotal. Therefore, at minimum, the authors should provide replicates for one of the pseudoknots.

Third, the authors write that within one of the categories, the structural differences between predictions were minor and therefore only one structure was chosen for MD simulations. What is the criterium for deciding that one starting structure suffices? Are the conformation of the other predictions seen in the simulation started from the one predicted structure?

Do simulations started from different predicted structures give rise to the same structural ensemble as would be expected for equilibrated simulations?

Typo on P. 19: "a high level sequence of sequence identity"

Reviewer #4: In “Modelling the structures of frameshift-stimulatory pseudoknots from representative bat coronaviruses,” Sekar et al. build all-atom models of four diverse bat coronavirus frameshift elements using RNA secondary structure prediction, 3D modeling with FARFAR, and molecular dynamics with AMBER. The four bat models are compared with each other and also compared to the experimentally-determined structure for SARS-CoV-2. Overall, the modeling is interesting as a way to generate hypotheses for the structures, but the hypotheses are untested.

Major Revisions:

The modeling software tools used are state-of-the-art, but the modeling effort did not take advantage of the evolutionary conservation of structure (i.e. the underlying homology of the sequences). The manuscript does compare the models, but it is not made clear if the differences are a result of the evolution of structure or whether the modeling effort failed on some level because the expected conservation was not used. For example, secondary structure prediction was performed using single sequences, which is known to be helpful but not perfectly accurate. The best practice would have been to test the secondary structure models using a sequence alignment across the bat coronavirus frameshift elements. For KF294282, a structure different from the other 3 was predicted, but the manuscript is not clear whether the sequence is capable of folding in the same structure. An example of an alignment for conservation analyses are found in related literature [JACS. 2021. 143:11404]. Covariations of base pairs should have been tested for significance [Nature Methods. 2017. 14:45]. In the absence of comparative analysis, I am skeptical of the quality of the secondary structure models.

Another aspect of concern is that many analyses are qualitative. An example of this is the comparison of molecular dynamics simulations. Single simulations were run for each 3D model with or without magnesium. The comparisons lack necessary controls because there is no estimation of the variability for simulations of the same coordinates. I expect to see multiple simulations for starting coordinates and solvent conditions if simulations are to be compared.

Minor, Required Revisions:

It is unclear how much sequence was used to generate the phylogenetic clusters in Figure 1. Was it the full length genomes or a region at the frameshifting site? And, if it is a region, how much was used? What alignment tool was used for the alignment?

For the secondary structure predictions, it is unclear what is meant by the “lowest-energy consensus pseudoknotted structure”. Does this mean that HotKnots and pKiss predicted the same structure in most cases or does it mean that the lowest energy of the two predictions was chosen?

For the simulations performed with magnesium, how were the six ions placed? I am concerned that 1 us simulations are inadequate for equilibration of magnesium ions. This is discussed, for example, in [JCTC. 2016. 12: 3370] and references therein.

For the selection of the FARFAR models to use for simulations, how were the models for subsequent analysis chosen from the set of alternative structures?

A number of statements are qualitative, but should be made quantitative. For example, differences between models were stated to be “relatively minor.” KU182958 was stated to be “quite similar” to SARS-CoV-2. Fluctuations of L3 are stated to be “quite low” (immediately following the Fig. 4 caption).

It would be helpful if Figure 2 defined the colors in relationship to S1, S2, and S3. The reader needs to consult Fig. S1 to find the definitions.

The results of the clustering of conformations from the simulations was not detailed. For each set of coordinates, how many clusters were identified? Importantly, do clusters appear at multiple timepoints across trajectories or do clusters tend to appear once? If clusters appear transiently and do not recur, I am concerned that the simulations are unconverged. Here, multiple trajectories would also be helpful for testing convergence; for converged simulations I would expect cluster members to appear in independent trajectories when the trajectories are clustered together.

Much of the text compares fluctuations across simulations (across models or between simulations with and without magnesium). With single simulations for each model and solvent, it is unknown whether these differences are significant.

The short stem-loop in a subset of KF294282 models was stated to dissociate. It would be helpful to know the timeframe of dissociation.

**Have the authors made all data and (if applicable) computational code underlying the findings in their manuscript fully available?**

Reviewer #1: Yes

Reviewer #2: Yes

Reviewer #3: Yes

Reviewer #4: **No: **The authors do not provide a sequence alignment. (This would be easy for them to provide.)

PLOS authors have the option to publish the peer review history of their article (what does this mean?). If published, this will include your full peer review and any attached files.

Reviewer #1: No

Reviewer #2: No

Reviewer #3: No

Reviewer #4: No
---

## [Decision Letter · Decision Letter 1]

18 Mar 2023

Dear Dr. Woodside,

Thank you very much for submitting your manuscript "Modelling the structures of frameshift-stimulatory pseudoknots from representative bat coronaviruses" for consideration at PLOS Computational Biology.

As with all papers reviewed by the journal, your manuscript was reviewed by members of the editorial board and by several independent reviewers. In light of the reviews (below this email), we would like to invite the resubmission of a significantly-revised version that takes into account the reviewers' comments.

Thank you for addressing the comments of the prior round. Reviewer 4 still raises many points and areas for improvement that would make the work more quantitative, and we agree. Please address these issues in your revision, as we believe they will significantly improve your paper.

We cannot make any decision about publication until we have seen the revised manuscript and your response to the reviewers' comments. Your revised manuscript is also likely to be sent to reviewers for further evaluation.

Sincerely,

Tamar Schlick

Academic Editor

PLOS Computational Biology

Nir Ben-Tal

Section Editor

PLOS Computational Biology

Thank you for addressing the comments of the prior round. Reviewer 4 still raises many points and areas for improvement that would make the work more quantitative, and we agree. Please address these issues in your revision, as we believe they will significantly improve your paper.

Reviewer's Responses to Questions

**Comments to the Authors:**

Reviewer #1: The authors have performed extensive revision and successfully addressed all my concerns with additional simulation replica and clarification on previous conclusions. The authors revisited the multiple sequence alignment and performed covariation analysis according to reviewers’ comments. RNA secondary structure predictions were examined with additional pseudoknot prediction softwares and more 3D structures were selected for MD simulations to validate the simulation reliability and confirm the structural features. The authors also performed simulations starting from the same RNA structure with different Mg2+ ions positions and updated the discussion. The links between pseudoknot sequences and structures are much clearer, and the MD simulation results are much more solid than what was presented initially.

Reviewer #2: The authors have done a great job addressing all my comments. The manuscript is much clearer now. Glad to see more extensive analysis of sequence alignment and structure prediction. Though this manuscript provides qualitative features rather than quantitative measurements of different coronavirus FSEs, it is still important for future antiviral therapeutic studies. Look forward to the authors' future work addressing interesting questions like the varying PRF inhibitors' effects.

Reviewer #3: All my concerns and questions were sufficiently addressed. From my side the article is ready for publication.

Reviewer #4: I appreciate that the authors made substantial efforts to address reviewers’ concerns. In particular, the repetition of simulations provides additional information to address concerns of convergence and repeatability. Additional details for the conservation of secondary structure are also important.

Overall, I appreciate the importance of the work. Excellent modeling methods were used. I am still somewhat concerned that the manuscript is qualitative in its description of the results. There are missed opportunities to be more quantitative. For example, given replicates of simulations, 2D RMSD plots would assess the similarity of conformations sampled across the structures. Additionally, the extent to which base pairs are preserved in the calculations could be quantified.

There are still some details that must be provided to fully evaluate and reproduce the work:

First, infernal is stated to be used to align the sequences. My understanding is that infernal requires a covariance model to function, therefore there must have been an alignment that was made first to train the covariance model. The detailed steps need to be clear. Even if there is a program in infernal that aligns de novo and I am wrong, infernal is a large collection of tools and therefore the details should be specified.

Figure S1 shows the dot-bracket representation of the structure, but does not show the nucleotide identities. The alignment should show both the nucleotides and the base pairing.

Six magnesium ions were added to each magnesium-containing simulation. The manuscript should specify how the exact location was chosen. Also, the manuscript might comment on the extent to which they move from their starting location during the simulations.

For the clustering of simulation snapshots, there are additional details that need to be provided. Please clarify whether the RMSDs were calculated using mass weighting, heavy atoms, or all-atoms with no weights. Please also specify if the clustering was done with multiple trajectories concatenated or whether each trajectory was clustered by itself. The supplementary figures color annotate the clusters as a function of time. The colors are repeated, so I can hypothesize the clusters are the same (from concatenating) or that the coloring is just repeated. (In my opinion, either approach is fine; I just want to know what was done.)

Minor suggestions:

The term “blind structure prediction” is used many times, starting in the abstract. It would be helpful if the manuscript specified to what the predictions are blind. By reading the manuscript, I understand it is blind to prior experimental structures, but this could be made explicit.

On page 4 (of the marked up version), “sharing also sharing” should be “also sharing”.

On page 7, the manuscript states that trajectories were tested to make sure there were not “large” changes in RMSD. The manuscript should quantify “large”. Also, in the same paragraph, the hydrogen bond distance criterion is specified. Was there also a criterion for the angle of the atoms? I am accustomed to seeing a geometry requirement as well, so if one was not used, it is probably best to be clear about that.

On page 9, the manuscript states “we also applied R-scape on the MSA to evaluate the conservation”. Although R-scape provides conservation information, its true value is in assessing the significance of covariations. The fact that two covariations were significant in S1 is very important support for the model. I suggest the manuscript be revised to explain this and emphasize the importance of the covariations.

On the bottom of page 9, the caveat that infernal does not handle pseudoknots is stated. I think it would be better to state this limitation up front when the manuscript states that infernal was used.

Figures 6, 7, 8, and 9 state that panel A is a structure with magnesium and panel B is a structure without magnesium. These must be backwards. I assume the green spheres are magnesium ions. (And starting in Figure 3, it would be helpful if the caption clarified that the green spheres are magnesium.)

In Figure 6, I noticed that two magnesiums intervene between S1 and S3, causing a disruption of a coaxial stack. The text comments on a base pair being disrupted in S3, but the loss of the coaxial stack seems like a more prominent change to me. It would be helpful if the text clarified whether this change occurs reliably across the two simulations.

In Figure 9, I note that, in the absence of magnesium, the central loop is quite open in a way that seems very wrong as compared to native structures. The manuscript refers to the tertiary interactions being sparse. The magnesium-containing structure is much more compact, as expected. It would be helpful if the manuscript stated whether this difference is reliable across the repeated simulations.

**Have the authors made all data and (if applicable) computational code underlying the findings in their manuscript fully available?**

Reviewer #1: Yes

Reviewer #2: Yes

Reviewer #3: Yes

Reviewer #4: Yes

PLOS authors have the option to publish the peer review history of their article (what does this mean?). If published, this will include your full peer review and any attached files.

Reviewer #1: No

Reviewer #2: No

Reviewer #3: No

Reviewer #4: No
---

## [Editor Report · Decision Letter 2]

24 Apr 2023

Dear Dr. Woodside,

We are pleased to inform you that your manuscript 'Modelling the structures of frameshift-stimulatory pseudoknots from representative bat coronaviruses' has been provisionally accepted for publication in PLOS Computational Biology.

Best regards,

Tamar Schlick

Academic Editor

PLOS Computational Biology

Nir Ben-Tal

Section Editor

PLOS Computational Biology

---

## [Editor Report · Acceptance letter]

15 May 2023

PCOMPBIOL-D-22-01169R2 

Modelling the structures of frameshift-stimulatory pseudoknots from representative bat coronaviruses

Dear Dr Woodside,

I am pleased to inform you that your manuscript has been formally accepted for publication in PLOS Computational Biology. Your manuscript is now with our production department and you will be notified of the publication date in due course.

With kind regards,

Anita Estes
